



# Measurement report: Size distributions of urban aerosols down to 1 nm from long-term measurements

Chenjuan Deng[1], Yiran Li[1], Chao Yan[2,3,4], Jin Wu[1], Runlong Cai[2], Dongbin Wang[1], Yongchun Liu[3], Juha Kangasluoma[2,3,5], Veli-Matti Kerminen[2], Markku Kulmala[2,3,4], Jingkun Jiang[1,*]

[1]State Key Joint Laboratory of Environment Simulation and Pollution Control, School of Environment, Tsinghua University, 100084 Beijing

[2]Institute for Atmospheric and Earth System Research/Physics, Faculty of Science, University of Helsinki, 00014 Helsinki, Finland

[3]Aerosol and Haze Laboratory, Beijing Advanced Innovation Center for Soft Matter Science and Engineering, Beijing University of Chemical Technology, 100029 Beijing, China

[4]Joint International Research Laboratory of Atmospheric and Earth System Sciences, School of Atmospheric Sciences, Nanjing University, Nanjing, China

[5]Karsa Ltd., A. I. Virtasen aukio 1, 00560 Helsinki, Finland

*Correspondence to: Jingkun Jiang (jiangjk@tsinghua.edu.cn)





**Abstract**. The size distributions of urban atmospheric aerosols convey important information on their origins and impacts. Their long-term characteristics, especially for sub-3 nm particles, are still limited. In this study, we examined the characteristics of atmospheric aerosol size distributions down to 1 nm based on four-year measurements in urban Beijing. Using cluster analysis, three typical types of number size distributions were identified, i.e., daytime new particle formation (NPF) type,

daytime non-NPF type, and nighttime type. Combining a power law distribution and multiple lognormal distributions can well represent the sharp concentration decrease of sub-3 nm particles with increasing size and the modal characteristics for those above 3 nm in the submicron size range. The daytime NPF type exhibits high concentrations of sub-3 nm aerosols together with other three modes. However, both the daytime non-NPF type and the nighttime type have a low abundance of sub-3 nm aerosol particles together with only two distinct modes. In urban Beijing, the concentration of $H_2SO_4$ monomer during the

daytime with NPF is similar to that during the daytime without NPF, while significantly higher than that during the nighttime. The concentration of atmospheric sub-3 nm particles on NPF days has a strong seasonality while their seasonality on non-NPF days is less pronounced. In addition to NPF as the most important source, we show that vehicles can emit sub-3 nm particles as well, although their influence on the measured aerosol population strongly depends on the distance from the road.




## 1 Introduction

Atmospheric aerosol particles originate from both primary and secondary sources, spreading a wide range of sizes from ~1 nm
to hundreds of micrometers. Primary sources directly emit particles over a broad size range, and these sources include natural
ones such as windblow dust and sea spray and anthropogenic ones such as coal combustion, traffic emissions, and biomass
burning. Secondary sources produce mainly fine and ultrafine particles (Kumar et al., 2014). Atmospheric new particle
formation (NPF), for example, generates a significant number of particles down to ~1 nm, which have been shown to be
ubiquitous in the atmosphere (Zhang et al., 2012). Traffic emissions can contain sub-3 nm particles and those particles are
detected when measurements are made near the sources (Ronkko et al., 2017). In urban atmospheric environment mixed with
various emissions, the size distributions of atmospheric aerosol particles are dynamic and complex, reflecting contributions
from many primary and secondary sources.

For decades of development, size distributions of atmospheric particles larger than 3 nm have been well understood in contrast
to limited information about sub-3 nm particles. Whitby tri-modal representation (Whitby, 1978) has been widely used to
describe atmospheric aerosol size distributions, i.e., nuclei mode (~5-100 nm), accumulation mode (~100 nm - 2 $\mu m$), and
coarse mode (>2 $\mu m$). These three modes reflect their origins. For instance, nuclei mode particles are considered to be mainly
originated from gas-to-particle conversion. As the understanding of nucleation process extends, a number of studies proposed
the presence of nucleation mode in the size range of 3-25 nm which overlaps with Whitby nuclei mode (Covert et al., 1996;
Hoppel and Frick, 1990; McMurry et al., 2000). For instance, both modeling and experimental results showed that nucleation
mode is present when NPF events occur in the atmosphere (McMurry et al., 2000; Dal Maso et al., 2005; Hussein et al., 2004).

Advancing measurement techniques enables the development of an improved picture of atmospheric aerosol size distributions.
Prior to Whitby tri-modal representation, the prevailed size distribution of atmospheric aerosols was the power function model
by Junge (Junge, 1963), which was based on measurements using impactors with a large cut-off size (~0.1 μm) and low size
resolution. Developing electrical mobility-based techniques provided data for the tri-modal representation by achieving size
distribution measurements down to tens of nanometers with high time resolution and high size resolution (Knutson and Whitby,
1975; Whitby and Clark, 1966). Electrical mobility size spectrometers are now widely used to measure atmospheric aerosol
size distributions. During the last decade or so, advanced techniques were developed and improved towards measuring sub-3
nm atmospheric aerosols, such as diethylene glycol based electrical mobility spectrometer (DEG SMPS) (Jiang et al., 2011a),
particle size magnifier (PSM) (Vanhanen et al., 2011), neutral cluster and air ion spectrometer (NAIS) (Mirme and Mirme,
2013), half-mini differential mobility particle sizer (Half-mini DMPS) (Kangasluoma et al., 2018) and differential mobility
analyzer train (DMA train) (Stolzenburg et al., 2017). Among them, DEG SMPS, PSM, and NAIS have been used in a number
of field measurements (Deng et al., 2021; Jiang et al., 2011b; Kontkanen et al., 2016; Kontkanen et al., 2017; Sulo et al., 2021).

These developments and applications improve the understanding about size distributions of atmospheric sub-3 nm particles.
Jiang et al. (2011b) first measured atmospheric size distributions down to ~1 nm using DEG SMPS during a short-term
campaign in Atlanta and showed the sharp concentration decrease of sub-3 nm particles. The measured aerosol size distribution
was further verified by simulation and observation in chamber experiments with sulfuric acid and amine clustering conditions
(Chen et al., 2012). Studies measured the concentration of sub-3 nm particles using the PSM at various sites from a boreal
forest to polluted megacities (Kontkanen et al., 2017; Kontkanen et al., 2016). Despite these progresses, the characteristics of
sub-3 nm particles are still limited. For instance, whether sub-2 nm aerosols always exist with high concentrations in the
atmosphere is uncertain, although it is generally agreed that ion clusters in this size range are constantly present. DEG SMPS
measurements report high concentrations of sub-2 nm aerosols only during NPF periods (Jiang et al., 2011b) while PSM
measurements observe sub-2 nm signals all the time and with elevated concentrations during NPF periods (Kontkanen et al.,
2016; Kontkanen et al., 2017). This discrepancy can be partly attributed to different principles of these two instruments
(Kangasluoma et al., 2020). Long-term measurements in various atmospheric environments together with improved



understanding of their origins will help to better address this. Additionally, extending Whitby tri-modal representation down to sub-3 nm will provide a full picture of atmospheric aerosol size distributions (Kulmala et al., 2022). More importantly, it can provide information on nucleation processes where 1-3 nm is the critical size range, which can contribute to mechanistic, regional, and global atmospheric models.

To reveal the characteristics of atmospheric aerosol size distributions down to ~1 nm, we started long-term atmospheric
measurements in urban Beijing since 2018. Key gaseous precursors for the formation of sub-3 nm particles such as sulfuric acid and its clusters are also measured. This study aims to investigate the characteristics of typical aerosol size distributions together with gaseous precursors from the long-term perspective, the representation of the aerosol size distribution down to ~1 nm, and the possible origins of sub-3 nm aerosols. Particularly, this study focuses on the sub-3 nm size range, including its origin and connection with the rest of the submicron size range, especially the nucleation mode and the nuclei mode.

**2 Methods**

**2.1 Measurements**

Atmospheric measurements were conducted at two urban sites in Beijing (Fig. 1). Long-term measurements were carried out on the west campus of Beijing University Chemical Technology (BUCT site) since Jan. 2018. This site is situated ~550 m to the west of the 3$^{rd}$ ring Road and ~130 m to the southwest of a road, which is likely influenced by traffic emissions (Lu et al.,
2019; Cai et al., 2021b; Deng et al., 2020). Due to instrument maintenance, the data used in this analysis including those from Jan. 16 – May 17 and Oct. 19 – Dec. 26, 2018, Jan. 1 – Mar. 28, 2019, and Jul. 19, 2019 – Dec. 31, 2021 (a total of 1009 available days). In addition, a short-term campaign was conducted on the campus of Tsinghua University (Tsinghua site) during Mar. 7 – Apr. 6, 2016 (a total of 31 available days). Different from the BUCT site, the Tsinghua site is considered to be less influenced by traffic emissions because the closest major road is ~1 km away from this site (Cai et al., 2017b; Cai and
Jiang, 2017). There are no significant stationary emission sources nearby both two sites. Details about these two sites can be found in previous studies (Deng et al., 2020; Cai et al., 2017b).

==================

Place Figure 1 Here

==================

Size distributions of atmospheric aerosols in the range of 1 nm – 10 μm (mobility diameter) were measured using a home-made DEG SMPS (1 – 6.5 nm) (Jiang et al., 2011a; Cai et al., 2017a) and a particle size distribution spectrometer (PSD; 3 nm – 10 μm) (Liu et al., 2016). The schematics and pictures of the DEG SMPS and PSD are shown in Fig. S1 in the Supplementary Information (SI).The DEG SMPS is equipped with a core sampling inlet (Fu et al., 2019) for improving their sampling efficiency, a soft X-ray neutralizer (TSI Inc., model 3088), a specially designed miniature cylindrical differential mobility
analyzer (Cai et al., 2017a; Cai et al., 2019) for classifying sub-10 nm particles, and a two-stage condensation particle counter which includes a modified DEG-based ultrafine CPC and a conventional CPC (TSI Inc., 3772). The PSD consists of an aerodynamic particle sizer (APS; TSI Inc., model 3321) and two parallel SMPSs using a nano-DMA (model 3085, TSI Inc.) and a long-DMA (model 3081, TSI Inc.), respectively.

Concentrations of sulfuric acid and its clusters were measured using a nitrate chemical ionization time-of-flight mass
spectrometers (Aerodyne Research Inc.) (Zheng et al., 2015; Cai et al., 2017b; Lu et al., 2019). The sampling configurations and calibration procedures were reported previously (Lu et al., 2019; Zheng et al., 2015). The NO concentration was measured by a trace gas analyzer (42i-TL, Thermo Fisher). The meteorological data, including the temperature, relative humidity and ambient pressure were measured using local weather station data acquisition system (Vaisala, AWS310).





**2.2 Data analysis**

Cluster analysis was used to identify typical atmospheric aerosol number size distributions during the measurement period at both sites. Details about this methodology (Beddows et al., 2009; Wegner et al., 2012) are given in the SI. We identified three typical types of aerosol number size distributions in urban Beijing together with $H_2SO_4$ and its clusters: C1, C2 and C3 types (Figs. S2 and S3). C1 type has high concentration of sub-3 nm aerosols and is mostly observed during NPF periods (e.g., 9-14) on NPF days. The other two types are mostly observed during non-NPF periods. Thus, they are referred as daytime NPF type, daytime non-NPF type, and nighttime type, respectively. Their characteristics will be further discussed in the following section. The measurement days were classified into NPF days and non-NPF days according to the criteria and examples reported previously (Deng et al., 2020).

A combination of power law and multiple lognormal distribution functions was used to fit the measured $H_2SO_4$ and its clusters and particle number size distributions. When focusing on number size distributions, we fitted only in the submicron size range, because number concentration of coarse mode particles (>2 μm) is comparatively negligible. Note that to be consistent, we used mass diameter when combining size distributions of $H_2SO_4$ clusters and particles, otherwise mobility diameter is used for particles. The relation between mobility diameter and mass diameter (Ku and de la Mora, 2009), i.e., mobility diameter is 0.3 nm larger than mass diameter, was used to convert mobility diameter for particles into mass diameter.

For sub-3 nm particles, a power law function is used,

$$\frac{\mathrm{d}N}{\mathrm{dlog}d_p} = a d_{\mathrm{p}}^{-b} \qquad (1)$$

where $d_\mathrm{p}$ is the particle diameter, nm; $\frac{\mathrm{d}N}{\mathrm{dlog}d_p}$ is the number size distribution function, cm$^{-3}$; $a$ and $b$ are two fitting parameters for the power law function.

For particles above ~3 nm, multiple lognormal distribution functions (Seinfeld and Pandis, 2008) are used:

$$\frac{\mathrm{d}N}{\mathrm{dlog}d_\mathrm{p}} = \sum_{i=1}^{n} \frac{N_i}{\sqrt{2\pi}\log\sigma_{g,i}} \exp\left[ -\frac{(\log d_\mathrm{p} - \log\overline{d}_{pg,i})^2}{2\log^2\sigma_{g,i}} \right] \qquad (2)$$

where $d_\mathrm{p}$ is the particle diameter, nm; $N_i$, $\overline{d}_{pg,i}$ and $\sigma_{g,i}$ represent total number concentration (cm$^{-3}$), geometric mean diameter (nm) and geometric standard deviation (dimensionless) within the mode $i$.

The intensity of NPF is characterized by the particle formation rate, which measures the growth flux through a certain particle size. A balance formula that enhances the evaluation of coagulation scavenging in the presence of high aerosol loadings was used in this study to evaluate the particle formation rate (Cai and Jiang, 2017),

$$
J_k = \frac{\mathrm{d}N_{[d_k,d_u)}}{\mathrm{d}t} + \sum_{d_g=d_k}^{d_{u-1}} \sum_{d_i=d_{\min}}^{+\infty} \beta_{(i,g)} N_{[d_i,d_{i+1})} N_{[d_g,d_{g+1})}
$$
$$
- \frac{1}{2} \sum_{d_g=d_{\min}}^{d_{u-1}} \sum_{d_i^3=\max\left(d_{\min}^3, d_k^3-d_{\min}^3\right)}^{d_{i+1}^3+d_{g+1}^3 \leq d_u^3} \beta_{(i,g)} N_{[d_i,d_{i+1})} N_{[d_g,d_{g+1})} + \frac{\mathrm{d}N}{\mathrm{d}d_i}\bigg|_{d_i=d_u} \cdot \mathrm{GR}_u \qquad (3)
$$

Here $J_k$ is the particle formation rate at size $d_k$, cm$^3 \cdot s^{-1}$, where $d_k$ was 1.5 nm in this study, nm; $d_u$ is the upper size bound of the chosen aerosol population, nm; $d_{\min}$ is the smallest particle size detected by particle size spectrometers, nm; $N_{[d_k,d_u)}$ is the number concentration of particles from size $d_k$ to $d_u$, cm$^{-3}$; $d_i$ denotes the lower bound of the i[th] size bin, nm; $\beta_{(i,g)}$ is the coagulation coefficient for the collision of two particles with size of $d_i$ and $d_g$, cm$^3 \cdot s^{-1}$; and $\mathrm{GR}_u$ represents the particle growth rate at size $d_u$, nm·h$^{-1}$.



An indicator, $I$, represents the intensity of the NPF in atmospheric environment governed by $H_2SO_4$-amine nucleation, considering the effects of various parameters such as $H_2SO_4$ concentration, amine concentration, the stability of $H_2SO_4$-amine clusters, and background aerosols (Cai et al., 2021a). This indicator has been used to reveal the governing factor for the seasonal variations of NPF in urban Beijing (Deng et al., 2020). It was calculated using the method by Cai et al. (2021a) (details can be found in the SI). Together with other evidence, it is used to explore whether the elevation of sub-3 nm aerosols on non-NPF days is due to nucleation process in Sect. 3.2.

## 3 Results and discussion

### 3.1 Typical number size distributions of sub-3 nm aerosols

Figure 2 showed three cases of typical types of aerosol number size distributions. The number size distribution function ($dN/dlogd_p$) was at high levels in the sub-3 nm size range of the daytime NPF type size distribution, and the highest value in this case reached up to $\sim1.1\times10^6$ cm$^{-3}$. The $dN/dlogd_p$ decreased from $H_2SO_4$ monomer, $H_2SO_4$ dimer to sub-3 nm aerosols, and reached a trough at $\sim3$ nm (note that $H_2SO_4$ monomer and dimer are included in the size distribution to provide information about the precursors of nucleation process). This trough can be partly caused by aerosol dynamic processes, which have also been simulated by models (Chen et al., 2012; Li and Cai, 2020). After reaching the trough, the $dN/dlogd_p$ slightly increased but then dropped significantly. The $dN/dlogd_p$ showed no significant increase or decrease from $\sim12$ nm until it started to decrease substantially at $\sim 300$ nm and reached a low level at 1000 nm. In contrast to the daytime NPF type size distributions, the values of $dN/dlogd_p$ in the sub-3 nm size range of the daytime non-NPF type and nighttime type size distributions were substantially low. The concentrations of sub-2 nm aerosol particles were near zero. In the 2-3 nm size range, the $dN/dlogd_p$ of both daytime non-NPF and nighttime types was low while the $dN/dlogd_p$ was high in the larger size range, and higher than that of the daytime NPF type at sizes larger than about 20 nm. Although the daytime non-NPF type and nighttime type showed similar characteristics in the size range above the $H_2SO_4$ clusters, the $H_2SO_4$ monomer concentration was higher during daytime than nighttime.

==================

Place Figure 2 Here

==================

From a long-term perspective, the characteristics of the median aerosol number size distributions for each type were similar in four seasons (Fig. S4). For the daytime NPF type, they showed a similarly sharp particle concentration decrease with increasing size in the sub-3 nm size range and local peaks above $\sim3$ nm in four seasons, although the concentration level showed seasonal variations. The seasonal variations of number concentrations of sub-3 nm aerosols will be discussed in the following section. For the daytime non-NPF and nighttime type, both the pattern and concentration level were similar in four seasons.

The median atmospheric aerosol size distributions for the whole measurement time period can be generally well fitted using a combination function of the power law and the lognormal distributions (Fig. 3 and Figs. S5-6). Figure 3 showed that such combination function generally fitted the median daytime NPF type aerosol size distribution for the four-year measurements well, both in logarithmic and linear scale of ordinate. In Fig. 3(a), the power function depicted the characteristics that the concentrations decrease from $H_2SO_4$ monomers to $\sim3$ nm particles. Meanwhile, the lognormal function agreed well with the raw distributions. Figure 3(b) showed the tri-modal lognormal distributions more clearly in the linear scale of ordinate. The daytime NPF type size distribution presented three modes, referred as mode 1, mode 2, and mode 3, respectively. Unlike the



daytime NPF type, the daytime non-NPF and nighttime types only showed two modes in the number size distributions (Fig. S5-6).

================

Place Figure 3 Here


================

Table 1 summarized the fitted functions and their parameters of those median aerosol number size distributions for the four years of measurements in urban Beijing. The combination function consists of the power function and the lognormal distributions. For the power function, the parameters determining the fitting shape of sub-3 nm size distribution are $a$ and $b$. With a larger $b$ value, the concentrations decrease going from $H_2SO_4$ monomer to small clusters and further to particles of a 

few nanometers is shaper, so $b$ values were much larger for the daytime non-NPF and nighttime types than the daytime NPF type. For the daytime non-NPF and nighttime types, although their concentration in sub-3 nm size range were similarly low, the concentration decrease from $H_2SO_4$ monomer to dimer was sharper for the nighttime type, so the $b$ value was larger for the nighttime type than for the daytime non-NPF type. Concerning the lognormal distributions, the number concentration of mode 1 and mode 2 particles were much higher than that of mode 3 for the daytime NPF type. The modal diameters of mode 2 and 

mode 3 for the daytime non-NPF type and nighttime type were larger than those for the daytime NPF type.

================

Place Table 1 Here

================

Compared to the Whitby model, our results showed that in Beijing for the daytime NPF type number size distributions, there 

existed three modes (mode 1, mode 2, and mode 3) in the ultrafine size range. The first two modes are induced by atmospheric nucleation process and they are in the size range of nucleation mode. There was a trough between mode 1 and mode 2 in the daytime NPF type. This is because the nucleation process produces large numbers of small particles and only a fraction of them grows into larger sizes, together with a strong impact of coagulation in the small size range. Mode 3 in the daytime NPF type is mainly from primary emissions (Morawska et al., 1998; Ristovski et al., 1998), which is often called as Aitken mode.

In contrast to daytime NPF type, for the daytime non-NPF and nighttime type, there were only two modes (mode 2 and mode 3) in the ultrafine size range. The modal diameter in mode 2 for the daytime non-NPF and nighttime types was larger than the daytime NPF type. Also, for daytime non-NPF and nighttime types, the range of mode 2 was much broader and the number concentration in this mode was lower than that for the daytime NPF type. These characteristics indicate no influence of atmospheric nucleation process. Note that these fitting parameters are not presumed to be constant because atmospheric 

processes and emissions vary spatially and temporally.

The accumulation mode (100 nm – 2 μm) was not shown for all the types of number size distributions in Fig. 3 and Figs. S5-6 because their contribution in number concentrations was low. However, the accumulation mode obviously presented in the volume size distributions of all the types, and the concentration was lower for the daytime NPF type compared to other two types (Fig. S7). The accumulation mode particles showed high concentration in the surface area size distributions as well. This 

indicates their important role for scavenging smaller particles and condensing species.

Additionally, we showed that this combination model performs well in the measured aerosol size distributions in Atlanta (Fig. S8). The combination model captured the sharp particle concentration decrease with an increasing size in the sub-3 nm size range and the modal characteristics above 3 nm size of the daytime NPF type aerosol size distribution in Atlanta.



### 3.2 Potential sources of sub-3 nm aerosols in urban Beijing

NPF process is the most important source of atmospheric sub-3 nm particles ($N_{sub-3}$) in urban Beijing. As shown in Fig. 4(a), $N_{sub-3}$ showed a clear diurnal variation that reached its daily maximum in the noontime on NPF days. The median daily maximum $N_{sub-3}$ was ~$1.1 \times 10^4$ cm$^{-3}$ on NPF days in urban Beijing. The observed daytime maximum of $N_{sub-3}$ results mainly from the formation of the $H_2SO_4$ clusters through photochemical process during the daytime on NPF days (Fig. 4(c) and (d)). When the NPF events occurred, $N_{sub-3}$ was overwhelmingly higher than that when there were no NPF events (non-NPF daytime

or nighttime). Similar diurnal cycles of $N_{sub-3}$ and the dominant contribution of NPF process to the sub-3 nm aerosols were also observed in Shanghai, China (Xiao et al., 2015), Po Valley, Italy (Kontkanen et al., 2016), Kent, US (Yu et al., 2014), which are relatively polluted atmospheric environments.

===================

Place Figure 4 Here


===================

The concentrations of sub-3 nm aerosols showed a strong seasonality on NPF days but had no obvious seasonal variations on non-NPF days (Fig. 5 and Fig. S9), supporting that the elevated $N_{sub-3}$ was introduced by the NPF process. $N_{sub-3}$ was significantly higher in winter than those in summer on NPF days. The median daytime $N_{sub-3}$ on NPF days was ~$1.2 \times 10^4$ cm$^{-3}$ in winter in contrast to ~200 cm$^{-3}$ in summer. However, the more important reason is the much lower NPF intensity in summer

because the seasonal variation of $N_{sub-3}$ on NPF days was consistent with those of $H_2SO_4$ dimer concentration and the formation rates of ~1.5 nm aerosols (Fig. S10). This indicates that the seasonal variation of the formation process drives the seasonal variation of $N_{sub-3}$. This seasonal variation is different from that observed in Hyytiälä, a boreal forest site in Finland, in which $N_{sub-3}$ was the highest in summer or spring and the lowest in winter (Sulo et al., 2021). The highest concentration of small particles in the size range of 1.1 – 1.7 nm in Hyytiälä was observed during summertime, coinciding with the high

photochemical and biogenic activity in summer (Sulo et al., 2021).

===================

Place Figure 5 Here

===================

In addition to atmospheric NPF, sub-3 nm aerosols can also be emitted by primary sources, such as traffic emissions. This is

relatively easier to explore during non-NPF days. As shown in Fig. 4(a), $N_{sub-3}$ showed small morning (~6:00-9:00) and evening (~16:00-19:00) peaks on non-NPF days, roughly corresponding to the traffic rush hours in the morning and evening in urban Beijing. Although number concentrations of sub-2 nm aerosols were very low, those of 2-3 nm aerosols ($N_{2-3}$) showed diurnal patterns on non-NPF days (Fig. 4(b)), indicating that the morning and evening peaks are more likely due to primary emissions.

The obvious difference of the $N_{2-3}$ on non-NPF days between the Tsinghua site and the BUCT site, and between the COVID-19

lockdown period and normal period, further supports that vehicles emit 2-3 nm particles (Fig. 6). The morning and evening peaks of $N_{2-3}$ on non-NPF days were more prominent at the BUCT site (closer to traffic roads) than at the Tsinghua site that is considered to be less influenced by traffic emissions. In addition, we found that $N_{2-3}$ was much lower during the strict COVID-19 lockdown period at the BUCT site than normal time period. Traffic flows were extremely low during this strict lockdown period, such that traffic emissions were significantly reduced.

A non-NPF case (Mar. 13, 2018) was examined to further confirm that vehicles can emit 2-3 nm aerosols (Fig. 6(b) and (c)). On this non-NPF day, $N_{2-3}$ started to increase at around 6:00 and reached the maximum at around 8:00 when the concentration of the gas tracer for the traffic emissions, NO, also showed a peak. Furthermore, the NPF indicator, $I$, had low values during



this morning time, indicating that the increase of 2-3 nm aerosols was unlikely due to nucleation process (Fig. 6(c)). Previous studies reported that traffic can directly emit sub-3 nm aerosols and thus can be an important source for sub-3 nm aerosols
(Ronkko et al., 2017). Our results in urban Beijing support that traffic can emit 2-3 nm particles, but their relative contribution to the total aerosol number is negligible on NPF days.

==================

Place Figure 6 Here

==================

Figure 6 also showed that, unlike secondary formation process, traffic emissions in terms of sub-3 nm aerosols is of local characteristic and its impact on the measured aerosol population strongly depends on the distance between the traffic road and the measurement site. Nucleation and subsequent growth processes exist not only in the atmosphere but also in the exit of pipelines of vehicles (Giechaskiel et al., 2014). In the large-scale atmosphere, the relative homogeneity of sufficient gaseous precursors for nucleation and growth processes enables the burst of sub-3 nm aerosols. The lifetime of sub-3 nm aerosols is
extremely short due to strong coagulation effects in urban Beijing (Deng et al., 2021). Thus, sub-3 nm particles directly emitted from traffic are abundant only if one directly measures near the exhaust (Ronkko et al., 2017). This is supported by the fact that significantly less sub-2 aerosols were observed than 2-3 nm aerosols during the traffic rush hour both at the BUCT site and the Tsinghua site, i.e., due to the higher loss rate of sub-2 nm aerosols than 2-3 nm aerosols.

**4 Implications**

This study describes and interprets aerosol size distributions down to ~1 nm in urban atmospheric environments. Based on the modified Whitby model, we introduce the simplification of aerosol size distributions in sub-3 nm sizes, i.e., the power function. This fitting function captures well the sharp concentration decrease of sub-3 nm particles with an increasing particle size. Although different nucleation mechanisms exist in different atmospheric environments (Sipila, 2010; Jokinen et al., 2018; Lehtipalo et al., 2018; Yao et al., 2018; Yan et al., 2021; Beck et al., 2022), this simplified representation can be applied
because concentrations often decrease with an increasing size when going from precursor molecules to small clusters and further to aerosol particle of a few nanometers (Jiang et al., 2011b; Chen et al., 2012; Kulmala et al., 2022).

The aerosol size distributions down to ~1 nm and their representations are important parameters which can contribute to mechanistic, regional and global atmospheric models. For instance, for large-scale or regional models, the aerosol size distributions are initial input parameters to estimate aerosol population or CCN budgets (von Salzen et al., 2000; Adams, 2002).
In some models, the aerosol module often takes the modal representation of aerosol size distributions as input to simulate the aerosol dynamics for better computational speed (Binkowski and Roselle, 2003; Vignati et al., 2004). In urban environments with great complexity of precursors and emission sources, the representation of aerosol size distributions could be adjusted according to our findings instead of using the traditional three lognormal modal aerosol size distributions. Another benefit is that size distributions ranging from molecular levels to the submicron size range can help to better simulate the NPF and
growth mechanisms. For instance, Zhao and coworkers (Zhao et al., 2020) used a comprehensive model to investigate the NPF mechanisms in the Amazon free troposphere. They used measured aerosol size distributions with a lower size limit of ~10 nm to compare to simulated ones. Since sub-3 nm is the critical size range where nucleation occurs, it is needed to cover this size range when comparing simulations with atmospheric measurements. This is more important for urban environment mixed with various processes and more complex than pristine environment.

Based on long-term observational results, we addressed whether high concentrations of sub-2 nm aerosols are always present in the atmosphere. We show that concentrations of atmospheric sub-2 nm aerosols are high when NPF events occur. However,





they are significantly lower during non-NPF periods (often not detected by DEG SMPS) compared to NPF periods even though the $H_2SO_4$ monomer concentration is often similar. Different from results measured by DEG SMPS, previous studies reported that high concentrations of atmospheric sub-2 nm aerosols measured by PSM are constantly present in the daytime

(Kangasluoma et al., 2020; Kulmala et al., 2022). Although the higher noise-to-signal ratio and higher detection sensitivity of PSM may partly contribute (Kangasluoma and Kontkanen, 2017), considering that there is always high concentration of ion clusters in the atmosphere, the high signal of PSM all the time may be because that it measures both aerosol particles and ion clusters (Kangasluoma et al., 2020; Kulmala et al., 2022).

Additionally, our study implies that although vehicles can be massive in megacities, their direct emissions of sub-3 nm aerosols

only influence within the vicinities of traffic roads rather than the large-scale atmosphere. There are studies indicating that vehicles can emit high concentrations of sub-3 nm aerosols as detected by PSM (Ronkko et al., 2017; Okuljar et al., 2021). Similar to previous studies showing that concentrations of ultrafine particles emitted by vehicles decrease significantly as the distance from the roads increases, due to the rapid dilution and strong aerosol dynamic processes such as coagulation and condensation in the atmosphere (Zhu et al., 2002). Our study indicates that the decreasing phenomenon is more significant

with respect to sub-3 nm aerosols. Thus, though 2-3 nm aerosols emitted from vehicles were detected by DEG SMPS at both sites in urban Beijing, emissions of sub-2 nm aerosols were barely observed. Future studies measuring the size distributions of sub-3 nm aerosols simultaneously at sites with different distances from traffic roads can be performed to further investigate the impacts of vehicle emissions on atmospheric sub-3 nm aerosols without measurement interferences from ion clusters.

## 5 Conclusions

In this study, we identify three typical types of number size distributions based on four-year measurements using cluster analysis, i.e., daytime NPF type, daytime non-NPF type, and nighttime type, and investigate their characteristics. The daytime NPF type exhibits high concentrations of sub-3 nm aerosols together with other three modes. The first two modes are induced by atmospheric nucleation process and the third mode in the daytime NPF type is mainly from primary emissions. However, both the daytime non-NPF type and the nighttime type have a low abundance of sub-3 nm aerosol particles together with only

two distinct modes because they have no influence of nucleation process. In urban Beijing, the concentration of $H_2SO_4$ monomer during the daytime with NPF is similar to that during the daytime without NPF, while significantly higher than that during the nighttime. We use a power law distribution and multiple lognormal distributions to represent the sharp concentration decrease of sub-3 nm particles with increasing size and the modal characteristics for those above 3 nm in the submicron size range. This fitting function also performs well in Atlanta. We show that NPF is the major source of sub-3 nm particles in urban

Beijing. In addition to NPF, we find that vehicles can also emit sub-3 nm particles, although their influence on the measured aerosol population strongly depends on the distance from the road.

## Data availability

The datasets for this study can be accessible via https://doi.org/10.5281/zenodo.6654175. The details are available upon request from the corresponding author.



**Supplement link:** the link to the supplement is available at…

**Author Contribution**

C.D. and J.J. designed the research; C.D., Y.L., J.W., C.Y., Y.L., M.K. and J.J. collected the data; C.D. and J.J. analyzed data with the help from R.C., D.W., Y.C., V.K. and M.K.; C.D. and J.J. wrote the paper with inputs from all co-authors.

**Competing interests**

The author declares no competing interests.

**Acknowledgement**

Financial support from the National Science Foundation of China (22188102 and 92044301) and Samsung PM$_{2.5}$ SRP are acknowledged. We also acknowledge the following projects: ACCC Flagship by the Academy of Finland (337549), Academy professorship by the Academy of Finland (302958), Academy of Finland projects (1325656, 311932, 316114, 332547, and
325647), "Quantifying carbon sink, CarbonSink+ and their interaction with air quality" INAR project by Jane and Aatos Erkko Foundation, European Research Council (ERC) project ATM-GTP (742206).

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



**Table 1. The fitting parameters for the median daytime NPF, daytime non-NPF and nighttime type aerosol size distributions of four-year measurements in urban Beijing by the combination of the power law function and the lognormal distribution function.** The power law function includes two parameters, *a* and *b*. The lognormal distribution
function includes the number concentration within the mode (*N*), the geometric mean diameter within the mode ($d_{pg}$), and the standard deviation ($\sigma_g$).

| Fitting function | $\dfrac{dN}{d\log d_p} = a d_p^{-b}$ | | $\dfrac{dN}{d\log d_p} = \sum\limits_{i=1}^{n} \dfrac{N_i}{\sqrt{2\pi}\log\sigma_{g,i}} \exp\left[-\dfrac{(\log d_p - \log \bar{d}_{pg,i})^2}{2\log^2 \sigma_{g,i}}\right]$ | | | | | | | | |
|---|---|---|---|---|---|---|---|---|---|---|---|
| Size range | Sub-3 nm | | 3-1000 nm | | | | | | | | |
| Parameters | *a* | *b* | Mode 1 | | | Mode 2 | | | Mode 3 | | |
| | | | $N$ (cm⁻³) | $\bar{d}_{pg}$ (nm) | $\sigma_g$ | $N$ (cm⁻³) | $\bar{d}_{pg}$ (nm) | $\sigma_g$ | $N$ (cm⁻³) | $\bar{d}_{pg}$ (nm) | $\sigma_g$ |
| Daytime NPF type | 1.3×10⁶ | 6.1 | 1.1×10⁴ | 3.7 | 1.5 | 1.8×10⁴ | 9.8 | 1.7 | 6715 | 40.7 | 2.2 |
| Daytime non-NPF type | 3.8×10⁴ | 12.2 | - | - | - | 9000 | 20.7 | 2.2 | 9900 | 91.7 | 2.0 |
| Nighttime type | 3303 | 16.2 | - | - | - | 8000 | 19.7 | 2.2 | 9000 | 85.0 | 2.0 |



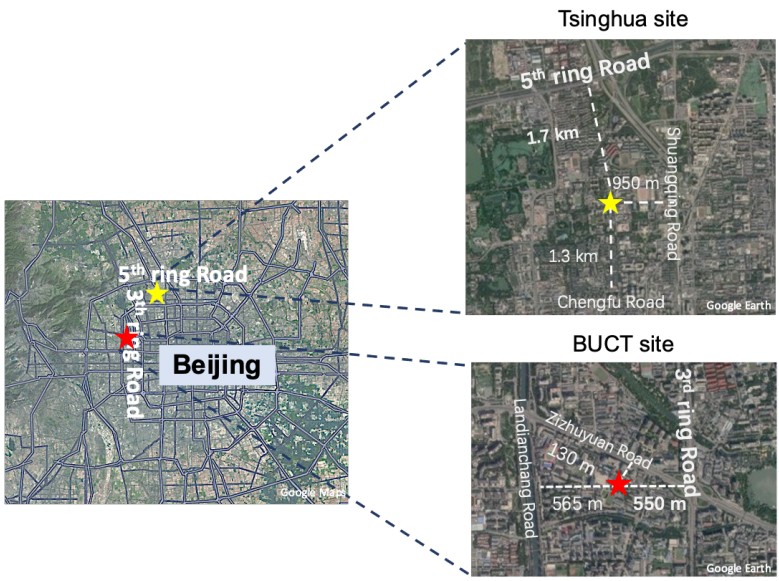


**Figure 1. The locations of two measurement sites in urban Beijing, i.e., Tsinghua site (yellow star) and BUCT site (red star).** The BUCT site is more influenced by the traffic emissions. The main roads and their distance from the measurement sites are marked in the figure. 5th ring road and 3rd ring road are the main roads near the Tsinghua site and the BUCT site, respectively. The maps are from © Google Maps and © Google Earth.






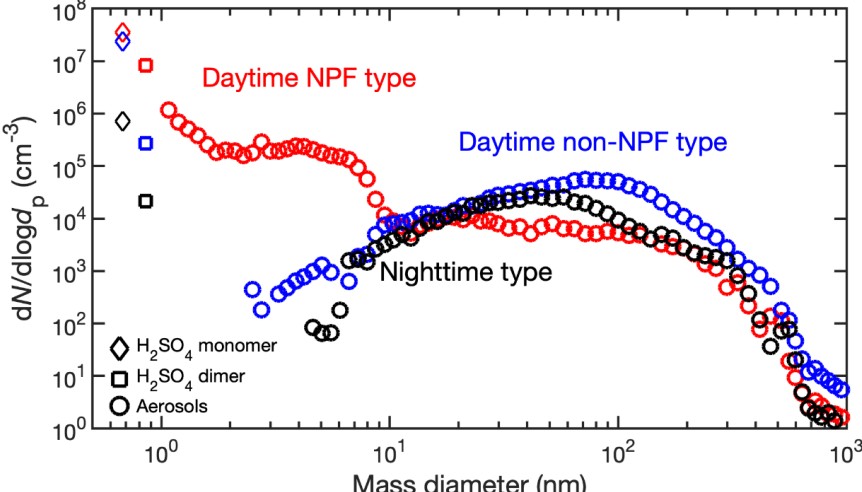

**Figure 2. Cases of three typical types of number size distributions from H₂SO₄ monomer to dimer and then to larger aerosol size: the daytime NPF type (red), daytime non-NPF type (blue) and nighttime type (black).** The diamonds, squares, and circles represent the distribution function (d$N$/dlog$d_p$) of H₂SO₄ monomers, dimers, and aerosols, respectively.






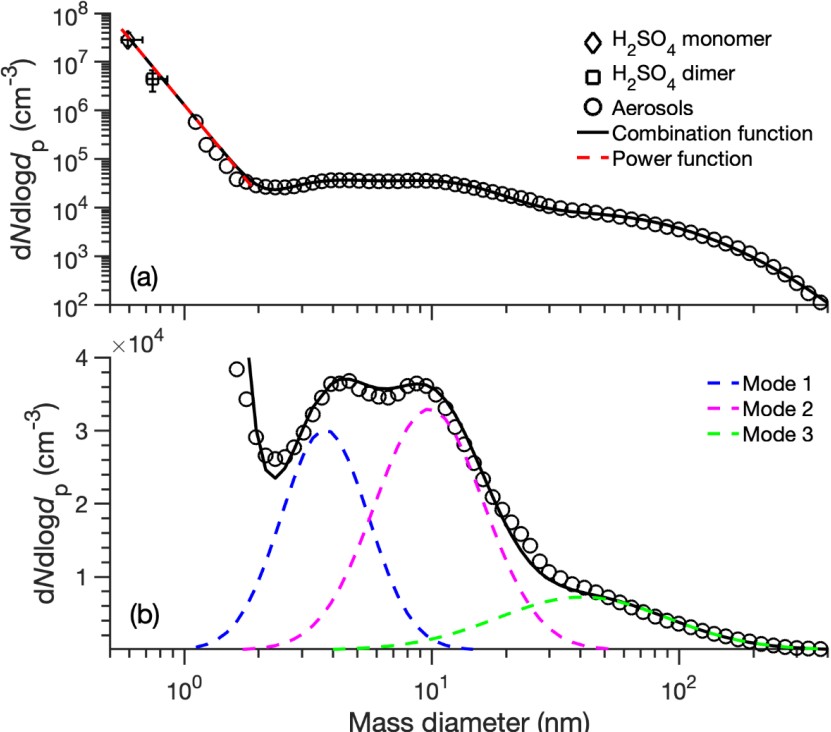

**Figure 3. The median daytime NPF type number size distributions from H₂SO₄ monomer to larger aerosols and the fitted size distributions shown in (a) logarithm scale and (b) linear scale of y axis.** The x-axis error bars of $H_2SO_4$ monomers and dimers indicate the variation range of estimated $H_2SO_4$ monomers and dimers diameters by assuming the bulk density to be 1000~1800 kg m³. The y-axis error bars of $H_2SO_4$ monomers and dimers indicate the 25th~75th range of concentrations. The black and red lines indicate the fitted size distribution in the whole size range and in sub-3 nm size range, respectively. The blue, magenta and green lines present the fitted mode 1, mode 2 and mode 3, respectively. The diamonds, squares, and circles represent the distribution function ($dN/dlogd_p$) of $H_2SO_4$ monomers, dimers and aerosols, respectively.

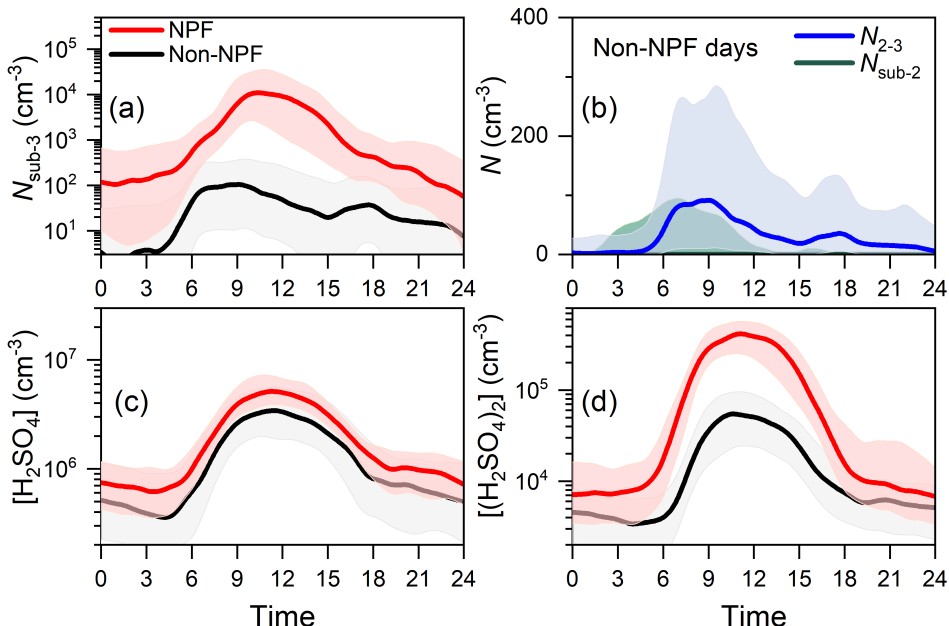


**Figure 4. The diurnal variations of (a) number concentration of sub-3 nm aerosols ($N_{sub-3}$), (c) H₂SO₄ monomer concentration and (d) H₂SO₄ dimer concentration on NPF days and non-NPF days. (b) The diurnal variations of number concentration of sub-2 nm aerosols ($N_{sub-2}$) and 2-3 nm aerosols ($N_{2-3}$) on non-NPF days.** The solid lines represent median values and the shading areas indicate $25^{th}$ – $75^{th}$ percentiles.


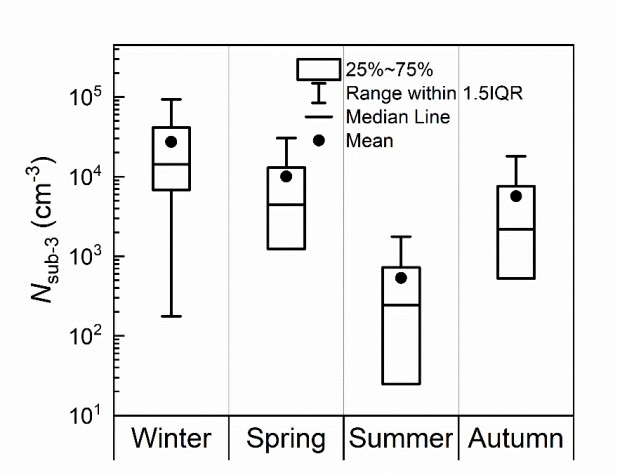

**Figure 5. The seasonal variations of number concentrations of sub-3 nm aerosols ($N_{sub-3}$) on NPF days in urban Beijing.**
Data during 9:00 – 14:00 were shown in this figure. The vertical lines and circles in the box indicate the median and mean
values, respectively. The top and bottom edges represent 75[th] and 25[th] percentiles, respectively. The IQR is the interquartile
range.





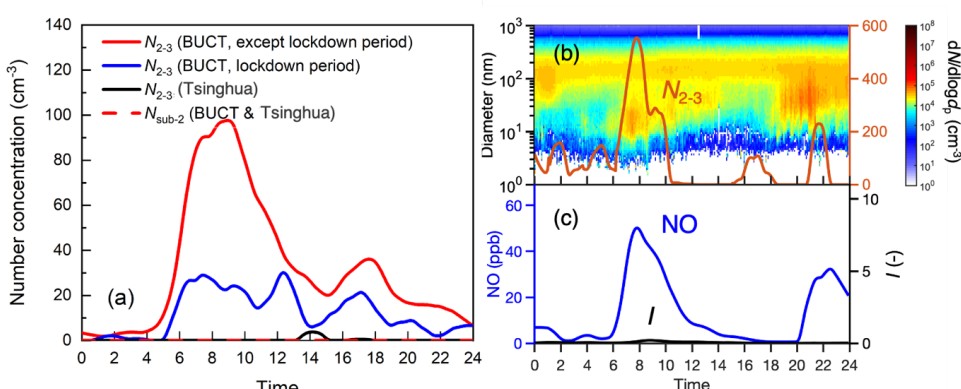

**Figure 6. (a) Median diurnal variations of number concentrations of 2-3 nm aerosols ($N_{2\text{-}3}$) on non-NPF days at the BUCT site during COVID-19 lockdown period (Jan. 24, 2020 – Feb. 25, 2020) and other time period (except lockdown period), and at the Tsinghua site (Mar. 7, 2016 – Apr. 6, 2016).** The median diurnal variations of number concentrations of sub-2 nm aerosols ($N_{\text{sub-2}}$) are near zero on non-NPF days at both BUCT and Tsinghua sites. The temporal pattern of (b) aerosol size distributions, $N_{2\text{-}3}$, (c) NO concentration and the indicator $I$ on a non-NPF day (Mar. 13, 2018).