# Peer review of "Measurement report: Size distributions of urban aerosols down to 1 nm from long-term measurements"

_Atmospheric Chemistry and Physics, 2022_

## Author Comment (AC1)

Measurement report: Size distributions of urban aerosols down to 1 nm from long-term measurements
Deng et al.
journal article response
en

**Responses to the Reviewer #1' Comments**

**"**Measurement report: Size distributions of urban aerosols down to 1 nm from long-term measurements" by Deng et al.

We appreciate valuable comments from the reviewer, which helped to improve this manuscript. We have addressed them in the following paragraphs (the text in italics is the comments, followed by our response). Additionally, all changes made are highlighted in the revised manuscript.

*This manuscript reports the characteristics of atmospheric aerosol size distributions from ~1 nm to 10 μm with an emphasis on sub-3nm particles from four-year measurements in urban Beijing. On the basis of cluster analysis, three typical types of number size distributions (i.e., daytime NPF type, daytime non-NPF type, and nighttime type) were identified. Based on the modified Whitby model, the simplification of aerosol size distributions in sub-3 nm sizes, i.e., the power function, was introduced. From the source identification of sub-3nm particles in urban Beijing, except for NPF, vehicle emission is another important source. Besides, the concentrations and diel patterns of $H_2SO_4$ monomer and dimer were also reported to better explain the formation mechanisms of NPF.*
*This is a unique dataset for which publication is worthwhile. This manuscript is well structured and written. Hence, I would ask one minor revision and recommend the publication of this article in Atmospheric Chemistry & Physics.*

*Minor comments:*

*Compared to lognormal distributions of larger particles, please add more discussion on why the power function was suitable to depict aerosol size distributions in sub-3 nm sizes.*
**Response:** Thanks for the suggestion. We have added some discussions on the reasons why the power function was chosen to depict aerosol size distribution in the sub-3 nm size range (lines 134-137 in the revised manuscript and Fig. S4 in the Supplementary Information (SI)):

"Note that the power law function was chosen to fit in the sub-3 nm size range because it can well capture the monotonic decrease from $H_2SO_4$ monomer to dimer and then to the sizes representative of aerosol particles. The log-normal distribution function is not a good fit in the sub-3 nm size range (Fig. S4), and especially a "mode" with a peak diameter of ~0.4 nm resulting from such a fit seems not to be reasonable."

*Figure 2, were $H_2SO_4$ monomer and dimer concentration converted into dN/dlogDp according to their mass diameters?*
**Response:** In Fig. 2 (and also Figs. 3, S3, S6, S7, and S9), we converted the $H_2SO_4$ monomer and dimer concentration into d$N$/dlog$d_p$ according to their mass diameters using the the method by Jiang et al. (2011). We added some descriptions of the conversion in the revised manuscript (lines 128-130):

"It should be also noted that, in Figs. 2, 3, S3, S6, S7 and S9, the concentration of $H_2SO_4$ monomers and dimers was converted into d$N$/dlog$d_p$ using the method by Jiang et al. (2011)."

*Figure 3, the label of Y-axis "dNdlogDp" should be "dN/dlogDp". Please check other figures in the*

*manuscript and SI.*

**Response:** Thanks for catching this. We have corrected it and carefully checked this issue throughout the entire manuscript and the SI.

*Figure 4, for panel C, if available, it would be better to add the diurnal patterns of solar radiation and $SO_2$. Also, around 5 a.m., the concentration of sulfuric acid started to rise. Did sulfuric acid follow the diel pattern of radiation at around 5 a.m.? If not, please comment on it.*

**Response:** We added a figure including the diurnal patterns of $H_2SO_4$ concentration, solar radiation, and $SO_2$ on NPF and non-NPF days during the measurement period in the SI (Fig. S10). As shown in Fig. R1 (Fig. S10 in the revised SI), the concentration of $H_2SO_4$ generally follows the diurnal pattern of solar radiation both on NPF and non-NPF days. This is because as the most vital oxidant to transfer $SO_2$ into $H_2SO_4$ by gas-phase oxidation, OH radicals are strongly correlated to solar radiation.

[Figure]

**Figure R1**. The diurnal variations of $H_2SO_4$ monomer concentration, $SO_2$ concentration and solar radiation on (a) NPF days and (b) non-NPF days during the measurement period. Median values were used in this figure.

However, $H_2SO_4$ concentration did increase at ~5 a.m. when there was basically no solar radiation both on NPF and non-NPF days. This might be induced by the production of OH radicals from the ozonolysis of alkenes during the nighttime. Previous studies indicate that the ozonolysis of alkenes can form OH radicals in the absence of solar radiation and thus can lead to the increase of $H_2SO_4$ concentration before sunrise (Guo et al., 2021). Other pathways forming OH radicals also likely exist for the increase of $H_2SO_4$ concentration before sunrise.

Even though there are pathways to produce $H_2SO_4$ without involving solar radiation, the photolysis of ozone is the most important source to produce OH radicals to form $H_2SO_4$ in our study. Also, the formation of $H_2SO_4$ is not the focus of our study, so we did not explain the possible formation pathways of $H_2SO_4$ in detail in the manuscript but added some discussions in the SI (lines 125-126):

"The increased $H_2SO_4$ concentration at ~5 a.m. might be induced by the production of OH radicals from the ozonolysis of alkenes during the nighttime (Guo et al., 2021)."

**Reference:**

Guo, Y., Yan, C., Li, C., Ma, W., Feng, Z., Zhou, Y., Lin, Z., Dada, L., Stolzenburg, D., Yin, R., Kontkanen, J., Daellenbach, K. R., Kangasluoma, J., Yao, L., Chu, B., Wang, Y., Cai, R., Bianchi, F., Liu, Y., and Kulmala, M.: Formation of nighttime sulfuric acid from the ozonolysis of alkenes in Beijing, Atmospheric Chemistry and Physics, 21, 5499-5511, 10.5194/acp-21-5499-2021, 2021.

Jiang, J., Zhao, J., Chen, M., Eisele, F. L., Scheckman, J., Williams, B. J., Kuang, C., and McMurry, P. H.: First Measurements of Neutral Atmospheric Cluster and 1–2 nm Particle Number Size Distributions During Nucleation Events, Aerosol Science and Technology, 45, ii-v, 10.1080/02786826.2010.546817, 2011.

---

## Author Comment (AC2)

**Responses to the Reviewer #2' Comments**

**"**Measurement report: Size distributions of urban aerosols down to 1 nm from long-term measurements" by Deng et al.

We appreciate valuable comments from the reviewer, which helped to improve this manuscript. We have addressed them in the following paragraphs (the text in italics is the comments, followed by our response). Additionally, all changes made are highlighted in the revised manuscript.

*I recommend the publication of this manuscript, but suggest the authors to address the following minor concerns.*

*1. line 120.*
*Your NPF definition looks like the classification of size distribution, not NPF. It's quite different from conventional NPF definition, like Dal Maso et al. Please describe the difference and how this will affect your conclusions.*

Response: We identified NPF and non-NPF using the classification method reported in Deng et al. (2020). According to this classification method, we identified a day with a burst of sub-3 nm particles and subsequent growth for hours as a NPF day and a day without burst of sub-3 nm particles and subsequent growth as a non-NPF day. A typical NPF and non-NPF day was shown in Fig. R2.

[Figure]

**Figure R2.** Typical particle size distributions of (a) a NPF day (Feb. 12, 2018) and (b) a non-NPF day (Nov. 26, 2018).

In Dal Maso et al. (2005), a day is identified into a NPF day if the nucleation mode (~3-25 nm) appears and those newly formed particles continue to grow over a time span of hours. The classification method used in our study is on the basis of the classification method in Dal Maso et al. (2005), but it can determine the starting point of the occurrence of NPF events more accurately with the information of sub-3 nm aerosols. Therefore, the difference between classification results by these two classification methods would be relatively small and would not affect the reported findings.

*2. line 129. why was power law function used? not other function? did you try any others? What information or contribution does the function provide on nucleation mechanistic, regional, and global atmospheric models?*

Response: We did try other function, e.g., the log-normal distribution function. As shown in Fig. R3, the size distribution of $H_2SO_4$ clusters and aerosols was fitted using the combination of four

lognormal distributions. In the sub-3 nm size range, the fitting is not so good and it fails to catch the rapid decrease of the distribution function from H$_2$SO$_4$ monomer to dimer and then to aerosol size because the lognormal distribution decreases slowly from the peak diameter to larger sizes. Furthermore, we would not think there is a "mode" with a peak diameter of ~0.4 nm existing in the sub-3 nm size range. Instead, the concentration decreases monotonically from gaseous precursors to aerosol size. This characteristic is well captured by the power law function; therefore, we choose it to fit the size distributions in the sub-3 nm size range.

[Figure]

**Figure R3.** The median daytime NPF type number size distributions from H$_2$SO$_4$ monomer to larger aerosols and the fitted size distributions using lognormal distributions.

We added the discussion on the reasons why we chose the power law function to fit the aerosol size distribution in sub-3 nm in the revised manuscript (lines 134-137 and Fig. S4):

"Note that the power law function was chosen to fit in the sub-3 nm size range because it can well capture the monotonic decrease from H$_2$SO$_4$ monomer to dimer and then to the sizes representative of aerosol particles. The log-normal distribution function is not a good fit in the sub-3 nm size range (Fig. S4), and especially a "mode" with a peak diameter of ~0.4 nm resulting from such a fit seems not to be reasonable."

As aforementioned, this power law function describes the pattern of size distributions in the sub-3 nm size range in a clear way and can contribute to modeling work. We also added the discussion on the contribution of the power law function to modeling studies (lines 300-306):

"Additionally, the power law function can be readily incorporated in models. For instance, in the global climate models, the observed aerosol size distributions are used to compare with simulated ones (Bergman et al., 2012) and the sub-3 nm size range is the key to simulate nucleation process accurately. However, due to relative scarcity of measured sub-3 nm particle size distributions around the world, the comparison between observed and simulated results is usually lacking this key size range. Our power law function, as the simplified representation of sub-3 nm size distributions, can extend the observed particle size distributions from above 3 nm to sub-3 nm, thus helping to constrain the aerosol module in global models."

*3. line 150-165. Is the PNSD for the time slot before, during or after NPF? how did you pick up the PNSD that you named "typical" from a large set of data? The PNSD is always changing even during a single NPF event.*

Response: Sorry for the confusion. The daytime NPF, daytime non-NPF and nighttime PNSD is selected during the NPF period on a NPF day (Feb. 16 11:10, 2018), during the daytime on a non-NPF day (Feb. 25 12:25, 2018) and during the nighttime (Apr. 4 00:35, 2018). We added this information in the caption of Fig. 2 (lines 541-543):

"The daytime NPF, daytime non-NPF and nighttime PNSD is selected during the NPF period on a NPF day (Feb. 16 11:10, 2018), during the daytime on a non-NPF day (Feb. 25 12:25, 2018) and during the nighttime (Apr. 4 00:35, 2018)."

The cluster analysis helped us to identify three types of PNSDs based on a large set of data, and then we picked up the PNSDs in Fig. 2 as typical types because they show pretty similar characteristics recognized by the cluster analysis. We agree that PNSD is always changing even during a single NPF event and we can never find two exactly same PNSDs. However, what we are trying to achieve is to reveal the common characteristics of the PNSDs based on a long-term dataset.

*4. figure 4b where is Nsub-2 curve?*

Response: In Fig. 4b, the median concentration of sub-2 nm aerosols is constantly near zero during the whole day. We added the notation about this issue in the caption of Fig. 4 (lines 564-565):

"Note that the median diurnal variations of $N_{sub-2}$ are near zero on non-NPF days in (b)."

*5. line 245-255. is there any evidence from nanoparticle/cluster chemical composition (e.g., $H_2SO_4$, amines, organics) measurement to support the source of these sub 3 nm particles were from traffic emission? for example, how were $H_2SO_4$ monomer and dimer observed in these traffic events?*

Response: It would be great if we could be able to identify the chemical composition of sub-3 nm particles. Unfortunately, to date, the lowest size limit of measuring chemical composition of atmospheric particles is ~5 nm in diameter by very few instruments, e.g., a thermal desorption chemical ionization mass spectrometer (TDCIMS) (Smith et al., 2004; Li et al., 2022). Further efforts are needed to extend the detection size limit for measuring chemical composition of atmospheric sub-3 nm particles.

Regarding the clusters we measured, they are majorly formed through secondary processes and are relatively abundant in the atmosphere even on non-NPF days, so the impacts of vehicles are difficult to identify. For example, on non-NPF days, the $H_2SO_4$ monomer and dimer concentration showed as similar diurnal variations as on NPF days, which peaked around noon (Fig. 4(c) and (d)). There was no exceptional increase of $H_2SO_4$ monomer and dimer concentration during traffic rush hours even though previous studies show that vehicles can emit primary $H_2SO_4$ that can be further converted to new particles through nucleation (Arnold et al., 2012; Ronkko et al., 2013). This indicates that $H_2SO_4$ monomers and dimers emitted by vehicles are negligible compared to those converted from photochemical oxidation in the large-scale atmosphere.

**Reference:**

Arnold, F., Pirjola, L., Rönkkö, T., Reichl, U., Schlager, H., Lähde, T., Heikkilä, J., and Keskinen, J.: First Online Measurements of Sulfuric Acid Gas in Modern Heavy-Duty Diesel Engine Exhaust: Implications for Nanoparticle Formation, Environmental Science & Technology, 46, 11227-11234, 10.1021/es302432s, 2012.

Bergman, T., Kerminen, V. M., Korhonen, H., Lehtinen, K. J., Makkonen, R., Arola, A., Mielonen, T., Romakkaniemi, S., Kulmala, M., and Kokkola, H.: Evaluation of the sectional aerosol microphysics module SALSA implementation in ECHAM5-HAM aerosol-climate model, Geoscientific Model Development, 5, 845-868, 10.5194/gmd-5-845-2012, 2012.

Dal Maso, M., Kulmala, M., Riipinen, I., Wagner, R., Hussein, T., Aalto, P. P., and Lehtinen, K. E.: Formation and growth of fresh atmospheric aerosols: eight years of aerosol size distribution data from SMEAR II, Hyytiala, Finland, Boreal Environment Research, 10, 323, 2005.

Deng, C., Fu, Y., Dada, L., Yan, C., Cai, R., Yang, D., Zhou, Y., Yin, R., Lu, Y., Li, X., Qiao, X., Fan, X., Nie, W., Kontkanen, J., Kangasluoma, J., Chu, B., Ding, A., Kerminen, V. M., Paasonen, P., Worsnop, D. R., Bianchi, F., Liu, Y., Zheng, J., Wang, L., Kulmala, M., and Jiang, J.: Seasonal Characteristics of New Particle Formation and Growth in Urban Beijing, Environ Sci Technol, 54, 8547-8557, 10.1021/acs.est.0c00808, 2020.

Li, X., Li, Y., Cai, R., Yan, C., Qiao, X., Guo, Y., Deng, C., Yin, R., Chen, Y., Li, Y., Yao, L., Sarnela, N., Zhang, Y., Petaja, T., Bianchi, F., Liu, Y., Kulmala, M., Hao, J., Smith, J. N., and Jiang, J.: Insufficient Condensable Organic Vapors Lead to Slow Growth of New Particles in an Urban Environment, Environ Sci Technol, 56, 9936-9946, 10.1021/acs.est.2c01566, 2022.

Ronkko, T., Lahde, T., Heikkila, J., Pirjola, L., Bauschke, U., Arnold, F., Schlager, H., Rothe, D., Yli-Ojanpera, J., and Keskinen, J.: Effects of gaseous sulphuric acid on diesel exhaust nanoparticle formation and characteristics, Environ Sci Technol, 47, 11882-11889, 10.1021/es402354y, 2013.

Smith, J. N., Moore, K. F., McMurry, P. H., and Eisele, F. L.: Atmospheric Measurements of Sub-20 nm Diameter Particle Chemical Composition by Thermal Desorption Chemical Ionization Mass Spectrometry, Aerosol Science and Technology, 38, 100-110, 10.1080/02786820490249036, 2004.